# Association between Survival and Time of On-Scene Resuscitation in Refractory Out-of-Hospital Cardiac Arrest: A Cross-Sectional Retrospective Study

**DOI:** 10.3390/ijerph18020496

**Published:** 2021-01-09

**Authors:** Hang A Park, Ki Ok Ahn, Eui Jung Lee, Ju Ok Park

**Affiliations:** 1Department of Emergency Medicine, Hallym University Dongtan Sacred Heart Hospital, Hwaseong-si 18450, Korea; hangapark@hallym.or.kr; 2Department of Epidemiology, School of Public Health, Seoul National University, Seoul 08826, Korea; 3Department of Emergency Medicine, Myongji Hospital, Hanyang University College of Medicine, Goyang-si 10475, Korea; arendt75@gmail.com; 4Department of Emergency Medicine, College of Medicine, Korea University, Seoul 02841, Korea; ironlyj@gmail.com

**Keywords:** refractory, out-of-hospital cardiac arrest, survival, scene time

## Abstract

It is estimated that over 60% of out-of-hospital cardiac arrest (OHCA) patients with a shockable rhythm are refractory to current treatment, never achieve return of spontaneous circulation, or die before they reach the hospital. Therefore, we aimed to identify whether field resuscitation time is associated with survival rate in refractory OHCA (rOHCA) with a shockable initial rhythm. This cross-sectional retrospective study extracted data of emergency medical service (EMS)-treated patients aged ≥ 15 years with OHCA of suspected cardiac etiology and shockable initial rhythm confirmed by EMS providers from the OHCA registry database of Korea. A multivariable logistic regression analysis was conducted for survival to discharge and good neurological outcomes in the scene time interval groups. The median scene time interval for the non-survival and survival to discharge patients were 16 (interquartile range (IQR) 13–21) minutes and 14 (IQR 12–16) minutes, respectively. In this study, for rOHCA patients with a shockable rhythm, continuing CPR for more than 15 min on the scene was associated with a decreased chance of survival and good neurological outcome. In particular, we found that in the patients whose transport time interval was >10 min, the longer scene time interval was negatively associated with the neurological outcome.

## 1. Introduction

Despite significant advances in resuscitation science and practice, survival following out-of-hospital cardiac arrest (OHCA) is still low. In Korea, the number of OHCA cases in 2016 was 29,832, and the rate of survival was 7.6% [1]. In the USA, OHCA accounts for over 350,000 deaths annually, and the rate of survival to hospital discharge was 10.5% in 2017 [2].

Ventricular fibrillation (VF) or ventricular tachycardia (VT) was monitored in 18.7% of the emergency medical services (EMS)-treated OHCA patients, using an automated external defibrillator (AED). The survival rate of these patients was 29.3% in 2017 in the US [2]. Shockable rhythms are known to be strongly associated with improved survival rates compared to non-shockable rhythms in OHCA [3,4]. Nevertheless, it is estimated that more than 60% of OHCA patients with shockable rhythm are refractory to resuscitation and fail to achieve prehospital return of spontaneous circulation (ROSC) [5].

In recent years, patients with refractory OHCA (rOHCA) have been transported to hospitals that can offer advanced therapies such as mechanical cardiopulmonary resuscitation (CPR), targeted temperature management (TTM), extracorporeal life support (ECLS) therapy, and early percutaneous coronary intervention (PCI) [6]. In addition, there is increasing evidence that it may be helpful to transfer carefully selected patients [7,8,9] to the hospital early, with continuous CPR for further treatment [10,11]. In one notable research study, a transport algorithm that allows rapid transfer of selected rOHCA patients from the field, while undergoing CPR, showed improved outcomes [12]. However, there are no guidelines provided by the American Heart Association (AHA) on the timing of the transport of rOHCA patients [13,14,15]. The European Resuscitation Council guidelines suggest that patients with VF, VT, or treatable causes may be considered for early transport, after 10 min of advanced life support (ALS); however, there is no clear guidance on the timing of the transport [16].

Some studies have investigated the association between scene time intervals (STIs) and survival in rOHCA patients. These studies have shown that long stays at the scene have a negative impact on outcomes [14,16,17]. These studies included rOHCA patients with both a shockable and non-shockable initial rhythm; however, the rOHCA patients with a shockable initial rhythm showed a high probability of survival. Unfortunately, advanced treatments require greater number of medical resources, and hence, are provided only to selected patients.

Therefore, we focused on rOHCA patients with shockable initial rhythm and aimed to identify whether the field resuscitation time is associated with survival rate. In addition, considering that the transport time to the hospital can influence the decision of early transport, the relationship between the field resuscitation time and survival according to the transport time was investigated.

## 2. Methods

### 2.1. Study Design and Data Collection

This was a cross-sectional retrospective study, which used data from the OHCA registry database of Korea. In 2014, the Korean Cardiac Arrest Resuscitation Consortium (KoCARC) was established as a voluntary nationwide consortium of 62 hospitals. It is a collaborative research network that supports various research in the field of OHCA resuscitation [18]. The KoCARC registry includes patients with OHCA transported to the participating emergency departments (ED) by the EMS. Data were collected in a standardized registry form and entered a web-based electronic database registry. A quality management committee was constituted to monitor the completeness and consistency of the mandatory data variables. The committee verified and provided periodic feedback on the data to each hospital coordinator. The KoCARC registry contains data on at least one-third of the EMS-treated OHCA patients based on national statistics obtained from the Korean Centers for Disease Control and Prevention.

### 2.2. Study Setting

In Korea, the EMS system is based within the fire department, and certified EMS providers can provide basic life and advanced airway support under direct medical advice. EMS providers have provided advanced cardiac life support in the designated regions (18 of the total 85 cities and metropolitans in Korea) [19], under the auspices of the prehospital advanced cardiac life support (ACLS) project since July 2015. Only specially trained EMS providers, under direct medical oversight, are permitted to administer intravenous medication in the field [20]. All direct medical directions are performed via the phone. Patients who agreed to “Do not attempt to resuscitate” or patients with obvious signs of death did not receive resuscitation. The National Field Management Guidelines for EMS follow the criteria of the AHA guidelines for not starting CPR [15]. All other OHCA patients who do not meet these criteria, are transported to the nearest ED, with ongoing CPR by the EMS providers. The EMS provider’s declaration of termination of resuscitation (TOR) is legally prohibited [21]. As doctors are not dispatched with EMS, death cannot be announced in the field. Unless there is obvious evidence of death, all patients must be transferred to the hospital where the doctor can declare the death. Therefore, to be precise, some patients who do not show definite ROSC are not transported before ROSC, but rather transported while ROSC is undetermined or dead. A mechanical CPR device should be available in the ambulance; however, it is not mandatory in Korea.

### 2.3. Study Population

Data of all patients with OHCA of suspected cardiac etiology, who were treated by the EMS providers between 1 December 2015 and 30 June 2019, and were aged ≥15 years, were extracted from the KoCARC registry. Although various rOHCA definitions have been used, the most widely used definition is a cardiac arrest that requires more than 10 min of resuscitation or three or more defibrillation attempts [6].

Therefore, patients with a shockable initial rhythm (ventricular fibrillation, pulseless ventricular tachycardia, unspecified shockable rhythm) confirmed by the EMS providers, those who have been shocked more than three times by the EMS providers, and those with scene times of more than 10 min were included. We excluded patients with unknown clinical outcomes or unknown and unclear time information, such as EMS notification or EMS arrival and departure time; we also excluded those who did not receive CPR and those whose cardiac arrest occurred in a primary clinic or ambulance.

### 2.4. Main Outcomes

The primary outcome of this study was survival at the time of discharge (survival discharge). The secondary outcome was good neurological outcome at hospital discharge, defined as cerebral performance category (CPC) 1 or 2.

### 2.5. Definition of Variables

We collected information on emergency calls to dispatch centers, and EMS arrival at the scene, EMS departure, and ED arrival times. EMS arrival at the scene was defined as the point at which the EMS contacted the patient. The “Response time interval (RTI)” was defined as the time from call acceptance at the dispatch center to the time of EMS arrival at the scene. The STI was the interval between EMS arrival at the scene and the ambulance departure from the scene. We also classified STI into two groups based on a 15-min interval by considering the intersection of the probability of survival discharge and cumulative incidence of achieving prehospital ROSC as <15 and ≥15 min. The “Transport time interval (TTI)” was defined as the time from the departure of the ambulance to hospital arrival. The “Total prehospital time (TPT)” was the time between receiving an emergency call from the dispatch center and arrival at the ED. ROSC was defined as evidence of a palpable pulse or a measurable blood pressure. The prehospital ROSC was considered as any ROSC achieved at any point during resuscitation, even if was not sustained until the patient arrived at the ED [22].

### 2.6. Statistical Analysis

Descriptive analysis was conducted to identify the potential risk factors for survival to discharge. Categorical variables were presented as numbers and percentages, and the Chi-squared test was used to test between two groups. Continuous variables were presented as means with standard deviations or medians with interquartile ranges (IQR) and Student’s t-test or the Mann–Whitney U test were used for analysis.

A multivariable logistic regression analysis was conducted for the outcomes and STI groups. After we tested goodness of fit for survival discharge model and good neurological outcome model, we selected age, sex, bystander CPR, arrest location, RTI, TTI, and airway type as covariates. Adjusted odds ratios (aORs) and 95% confidence intervals (CIs) on outcomes were calculated. In addition, we assessed the interaction effects of each independent variable by adding interaction terms in the models. All statistical analyses were performed using SAS software, version 9.4 (SAS Institute Inc., Cary, NC, USA).

## 3. Results

Out of the 1749 OHCA patients with a shockable initial rhythm in the study period, 376 had rOHCA. Among them, the data of 344 patients who met the inclusion criteria were analyzed in the study (Figure 1).

### 3.1. Baseline Characteristics and Clinical Outcomes

Baseline characteristics of the study patients are shown in Table 1. Patients who survived at discharge were significantly younger, with fewer shocks delivered, compared to the patients who did not survive. The median STI of the non-survival and survival to discharge patients were 16 min (IQR 13–21 min) and 14 min (IQR 12–16 min), respectively. RTI was significantly longer in the non-survival to discharge group; however, TTI was not significantly different between the survival and non-survival to discharge groups.

The time variables and clinical outcomes were compared according to whether prehospital spontaneous circulation was restored (Table 2). RTI were significantly shorter in the group of patients with prehospital ROSC (median: 7 min, IQR: 6–9 min), and STI was also shorter (median: 14 min, IQR: 12–18 min). However, TTI and TPT were not significantly different between the two groups. The median time interval from EMS arrival on the scene to ROSC was 12 min (IQR 9–16 min). The rates of survival to discharge and good neurological outcomes were higher in the prehospital ROSC group.

### 3.2. Association between STI and Clinical Outcomes

The cumulative number of prehospital ROSCs increased up to 23 min of STI, after which convergence occurred. The probability of survival discharge also decreased until 23 min (Figure 2).

After adjusting for age, sex, bystander CPR, location of arrest, RTI, TTI, prehospital airway type, and prehospital ROSC, STI ≥ 15 min was significantly associated with better survival to discharge and good neurological outcome (adjusted OR (aOR), 0.33 (95% CI, 0.17–0.65) and aOR, 0.43 (95% CI, 0.22–0.86), respectively) (Table 3). Age was found to be negatively associated with both survival to discharge and good neurological outcome.

### 3.3. Association between STI and Good Neurological Outcome Stratified by TTI

When we assessed the effects of each independent variable, there was no significant interactions in the survival to discharge model (Appendix A). However, in a good neurological outcome model, it was confirmed that the interaction between the TTI and STI groups was significant (*p* < 0.05) (Appendix A). Considering that the median TTI was 7 min in our study, TTI was divided into groups of ≤5, 5–10, and >10 min to identify the association between STI and a good neurological outcome after stratification by TTI.

As shown in Figure 3, while maintaining other covariates at a fixed value, we could see a 10% decrease in the odds of good neurological outcome in the group with STI ≥ 15 min compared to the reference group, when the TTI was >10 min (aOR, 0.10; 95% CI, 0.02–0.55).

## 4. Discussion

Using the data from a nationwide multicenter OHCA registry in Korea, this retrospective observational study showed an association between STI and outcome in rOHCA patients with a shockable rhythm. After adjusting for other covariates, staying at the scene for more than 15 min was associated with a lower probability of survival to hospital discharge and good neurological outcome. In addition, it was also shown that the association between scene time and good neurological outcome differed depending on the transport time.

In the case of rOHCA, continuous resuscitation is required, meaning that clinical decision-making is necessary before the critical CPR time point is missed. The outcome of patients after OHCA is poor and worsens after prolonged resuscitation [8]. The EMS should decide whether to continue CPR on site or transport the patient to an ED to consider other options such as ECLS. However, owing to the lack of accurate criteria for patient selection, the optimal time of transport is still unclear [12]. According to our study results, it is not recommended to sustain CPR for more than 15 min in the field, and these results are in line with the findings of previous studies [16,17]. Importantly, it has been shown that 90% of prehospital ROSC occurs within the first 15 min of EMS resuscitation [23]. Therefore, performing at least 15 min of resuscitation in the field means conducting prehospital ROSC for as long as possible before transporting, and staying in the field longer will only reduce the chances for hospital-based post-resuscitation care.

The results of our study indicate that the decision on the length of stay at a scene may vary depending on the TTI. As in previous studies [24,25], TTI did not appear to have a significant impact on survival. These results serve as evidence for bypassing the nearest hospitals and transporting OHCA patients to cardiac arrest centers (CACs). Moreover, when OHCA patients present with shockable rhythms, direct delivery to CACs is suggested [26]. However, according to the research of Park et al., the detrimental effect of a longer TTI on neurologic outcome were more pronounced in the short STI group than in the long STI group among all OHCA patients [27]. The researchers emphasized that TTI and STI should be considered when choosing hospitals for OHCA patients without prehospital ROSC.

Whether the EMS provider’s or the doctor’s decision on departure from the field were affected by the expected transport times has not been studied yet, an issue that we believe requires further research. However, in our study, we showed that when the TTI is > 10 min, the longer STI was negatively associated with the neurological outcome in rOHCA patients. According to our results, if EMS providers decide to transport an rOHCA patient to a CAC more than 10 min away where ECLS and TTM are possible, they may need to consider reducing the time of on-scene resuscitation. Since the TTI is determined according to geographic characteristics, it is not a modifiable factor; nevertheless, it should be taken into consideration. Therefore, our study may be a basis for considering the initiation of the transfer along with the expected transfer times.

The novelty of this study that sets it apart from previous studies on STI is that the association between STI and outcome was observed in all rOHCA patients. All our study participants were transported to the ED; in addition, patients with a high probability of survival were not selectively transported. We analyzed all patients regardless of prehospital ROSC; nevertheless, we showed that a long STI had a negative effect on the outcome. A previous study had already shown that OHCA patients with prehospital ROSC have higher probability of survival than those without prehospital ROSC [28]. When we reanalyzed the multivariable logistic regression model without prehospital ROSC patients, STI was not significantly associated with survival to discharge (aOR, 0.45; 95% CI, 0.18–1.16). However, STI ≥ 15 min was still significantly negatively associated with good neurological outcome (aOR, 0.38; 95% CI, 0.22–0.67) (Appendix A). In patients who do not have prehospital ROSC, reducing STI and rapidly transporting them to the ED for further treatment, such as ECLS, may perhaps be associated with good neurological outcome. In particular, it may be more appropriate to perform an analysis without excluding patients with ROSC, because our study was aimed at determining whether shorter STI for rOHCA patients is associated with more favorable outcomes than waiting for ROSC in the field.

According to the directed acyclic graph for epidemiological relationships, hospital-level treatments were not considered as confounding variables; therefore, we did not include these variables in the main analysis. There are no clear criteria for ECLS application; however, the time from arrest to the start of ECLS is one of the important predictors of ECLS success [29]. The short prehospital time may influence the physician’s decision to perform ECLS.

This study had a few limitations. First, we could not measure the quality of CPR during transport. In some cases, it is possible that compression was maintained during transport using a mechanical device, but this factor was not considered as a potential confounder in the analysis. However, prehospital mechanical CPR has previously been shown to not be associated with outcomes [30,31]. Second, the STI and TTI were divided and analyzed based on the distribution of our data. Therefore, the possibility of generalization of these results to other countries is not clear. Third, in some areas where ACLS was provided under the direction of a physician, the physician may have decided to choose the type of advanced airway and whether to provide only BLS and start transferring the patient depending on the situation and the skill level of EMS providers. However, physicians do not provide instructions on how to perform procedures; thus, direct medical advice may not affect the success of procedures performed by the EMS providers. Even if only BLS was provided without an advanced airway, it would not have had a significant difference on survival since a previous study found no difference in survival between basic and advanced airway management [32]. Finally, in the case of endotracheal intubation and prehospital ROSC, a wide confidence interval was observed. Although, the study involved long-term data collection from multiple centers, the number of patients was not sufficient. We believe that further studies with larger patient populations are needed in the future.

## 5. Conclusions

In conclusion, for rOHCA patients with a shockable rhythm, continuing CPR for more than 15 min on the scene was associated with a decreased chance of survival and good neurological outcome. In particular, we found that in the patients whose TTI was >10 min, the longer STI was negatively associated with the neurological outcome. Our study may be a basis for future studies that consider the initiation of the transfer along with the expected transfer times.

## Figures and Tables

**Figure 1 ijerph-18-00496-f001:**
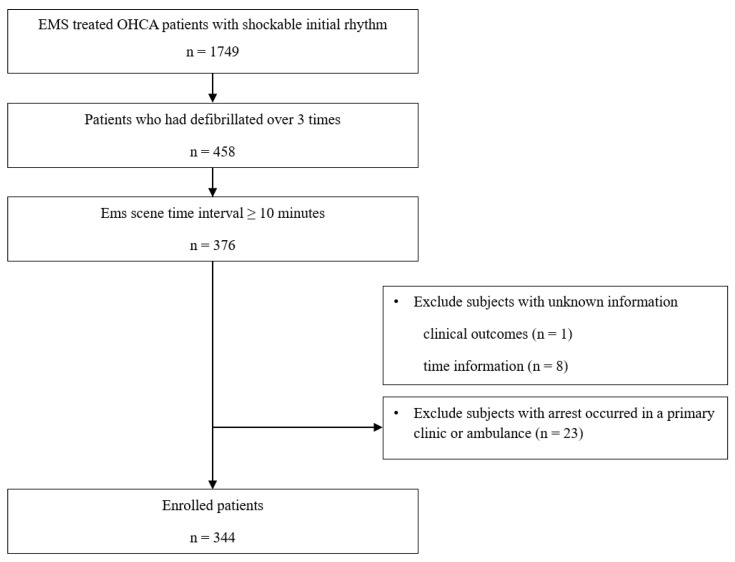
Flow chart showing patient recruitment according to the inclusion and exclusion criteria. Abbreviation: EMS, emergency medical service; OHCA, out-of-hospital arrest.

**Figure 2 ijerph-18-00496-f002:**
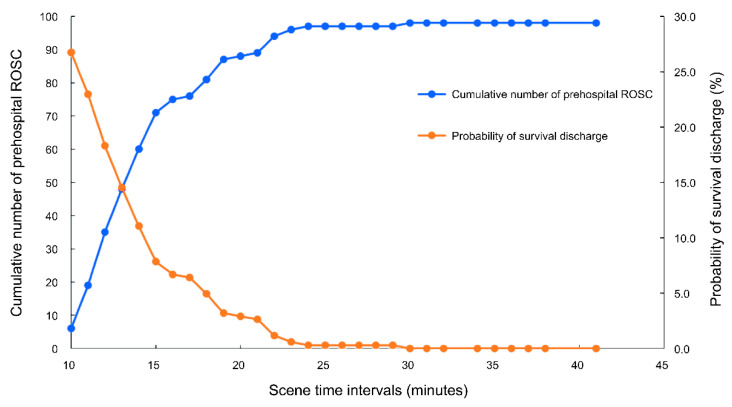
Cumulative number of cases with prehospital return of spontaneous circulation (ROSC) and probability of survival discharge according to scene time intervals.

**Figure 3 ijerph-18-00496-f003:**
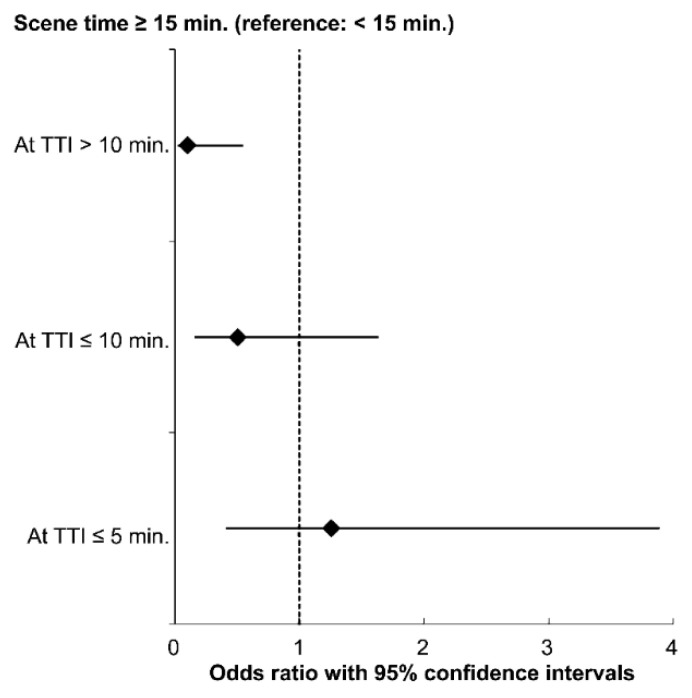
Multivariable logistic regression plot for good neurological outcome stratified by transport time intervals (TTIs). The models were adjusted for age, sex, bystander CPR, location of arrest, response time interval, prehospital airway type, prehospital ROSC achieved

**Table 1 ijerph-18-00496-t001:** Characteristics of patients by survival to discharge.

Variables	Total		Non-Survival Discharge		Survival Discharge		*p*
	*n*	%	*n*	%	*n*	%	
	344		246		98		
Age, years (mean, SD)	59.0	13.7	61.0	14.2	54.2	11.1	<0.001
Sex, male	283	82.3	202	82.1	81	82.7	0.91
Smoking status							<0.001
Current smoker	61	17.7	33	13.4	28	28.6	
Ex-smoker	23	6.7	12	4.9	11	11.2	
Never smoker	23	6.7	42	17.1	33	33.7	
Missing	185	53.8	159	64.6	26	26.5	
Hypertension	125	36.3	88	35.8	37	37.8	<0.001
Diabetes	60	17.4	46	18.7	14	14.3	<0.001
Time of event							0.87
07:00–19:00 (day)	192	55.8	138	56.1	54	55.1	
19:00–07:00 (night)	152	44.2	108	43.9	44	44.9	
Day of week							0.93
Weekday	227	66.0	162	65.9	65	66.3	
Weekend, holidays	117	34.0	84	34.1	33	33.7	
Witnessed (yes)	258	75.0	183	74.4	75	76.5	0.53
Bystander CPR (yes)	213	61.9	155	63.0	58	59.2	0.51
Arrest location							0.36
Non-public	175	50.9	129	52.4	46	46.9	
Public	169	49.1	117	47.6	52	53.1	
RTI, minutes	8	(6–10)	8	(6–10)	7	(6–8)	0.003
STI, minutes	15	(13–19)	16	(13–21)	14	(12–16)	<0.001
STI groups							<0.001
<15 min	151	43.9	91	37.0	60	61.2	
≥15 min	193	56.1	155	63.0	38	38.8	
TTI, minutes	7	(4–10)	7	(4–10)	7	(4–10)	0.90
TPT, minutes	31	(26–37)	32	(27–38)	29	(25–33)	<0.001
Number of shocks delivered	4	(3–6)	5	(4–6)	4	(3–5)	0.002
Airway types							0.003
Bag valve mask	80	23.3	53	21.5	27	27.6	
Supraglottic airway	230	66.9	176	71.5	54	55.1	
Endotracheal intubation	34	9.9	17	6.9	17	17.3	
Prehospital epinephrine given	84	24.4	73	29.7	11	11.2	<0.001
Prehospital amiodarone given	42	12.2	39	15.9	3	3.1	0.001
Prehospital ROSC	99	28.8	27	11.0	72	73.5	<0.001
ECLS applied	26	7.6	17	6.9	9	9.2	0.47
TTM applied	62	18.0	16	6.5	46	46.9	<0.001
Good neurological outcome	83	24.1	0	0.0	83	84.7	<0.001

Abbreviations: SD, standard deviation; CPR, cardiopulmonary resuscitation; RTI, response time interval; STI, scene time interval; TTI, transport time interval; TPT, total prehospital time; ROSC, return on spontaneous circulation; ECLS, extracorporeal life support; TTM, targeted temperature management. Categorical values were presented as numbers and percentage, continuous values as mean with standard deviation or median with interquartile range. The *p*-values were calculated using the chi-squared test for categorical variables and Student t-test or Mann–Whitney U test were used for continuous variables.

**Table 2 ijerph-18-00496-t002:** Time variables and clinical outcomes by prehospital ROSC.

Variables	Total		Prehospital ROSC		Without Prehospital ROSC		*p*
	344		99		245		
RTI, minutes	8	(6–10)	7	(6–9)	8	(6–10)	0.02
STI, minutes	15	(13–19)	14	(12–18)	16	(13–21)	0.001
STI groups							0.01
<15 min	151	43.9	55	55.6	96	39.2	
≥15 min	193	56.1	44	44.4	149	60.8	
TTI, minutes	7	(4–10)	8	(4–11)	6	(4–9)	0.07
TPT, minutes	31	(26–37)	29	(25–35)	32	(27–37)	0.07
EMS arrival to Prehospital ROSC			12	(9–16)			
Survival discharge	98	28.5	72	72.7	26	10.6	<0.001
Good neurological outcome	83	24.1	65	65.7	18	7.3	<0.001

Abbreviations: ROSC, return of spontaneous circulation; RTI, response time interval; STI, scene time interval; TTI, transport time interval; TPT, total prehospital time; EMS, emergency medical service. Time variables were reported as median and interquartile range and categorical variables as numbers and percentage. The *p*-values were calculated using the chi-squared test for categorical variables and Mann–Whitney U test were used for continuous variables.

**Table 3 ijerph-18-00496-t003:** Multivariate logistic regression analysis for outcomes.

Variables	Survival Discharge				Good Neurological Outcome			
	OR	95% CI	aOR	95% CI	OR	95% CI	aOR	95% CI
STI ≥ 15 min (<15 min)	0.37	(0.23–0.60)	0.33	(0.17–0.65)	0.42	(0.25–0.69)	0.43	(0.22–0.86)
Age, years	0.96	(0.95–0.98)	0.95	(0.93–0.97)	0.96	(0.94–0.98)	0.95	(0.93–0.98)
Male	1.04	(0.56–1.92)	0.78	(0.34–1.78)	0.97	(0.51–1.85)	0.69	(0.29–1.64)
Bystander CPR done	0.85	(0.53–1.37)	0.90	(0.47–1.72)	0.97	(0.59–1.62)	1.08	(0.55–2.13)
Public location (non-public)	1.25	(0.78–1.99)	0.77	(0.4–1.51)	1.49	(0.9–2.45)	1.14	(0.57–2.26)
RTI, minute	0.89	(0.81–0.97)	0.89	(0.78–1.00)	0.91	(0.83–0.99)	0.92	(0.81–1.04)
TTI, minute	1.03	(1.00–1.06)	1.02	(0.98–1.06)	1.03	(1.00–1.06)	1.02	(0.98–1.06)
Supraglottic airway (bag valve mask)	0.60	(0.35–1.05)	0.78	(0.37–1.65)	0.74	(0.41–1.34)	1.09	(0.49–2.41)
Endotracheal intubation (bag valve mask)	1.96	(0.87–4.44)	2.29	(0.74–7.11)	1.97	(0.85–4.58)	2.43	(0.76–7.80)
Prehospital ROSC	22.46	(12.32–40.96)	21.65	(11.19–41.90)	24.11	(12.79–45.47)	22.05	(11.19–43.48)

Abbreviations: OR, Odds ratio; CI, confidence interval; aOR, adjusted odds ratio; STI, scene time interval; RTI, response time interval; TTI, transport time interval; CPR, cardiopulmonary resuscitation; ROSC, return on spontaneous circulation. All references are shown in parentheses.

## Data Availability

The data presented in this study are available on request from the corresponding author. The data are not publicly available due to distribution of data is determined after the KoCARC research committee deliberation.

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
