# Peer review of "Association between Survival and Time of On-Scene Resuscitation in Refractory Out-of-Hospital Cardiac Arrest: A Cross-Sectional Retrospective Study"

_ijerph, 2021, doi:10.3390/ijerph18020496_

Round 1

Reviewer 1 Report

II congratulate you and your team on this article. I consider this to be an excellent contribution. It has managed to describe complex aspects with some clarity and this article is easily understandable considering the complexity of the matter.

If you allow me, I would like to make some comments to contribute, if you consider it, to its improvement.

In the "study setting" you say: "The EMS system is fire department based, and certified EMS providers can provide basic and advanced life support in the airways under direct medical advice." Can you explain what "direct medical advice and support" means? Is it by phone? Do you think that if they provided basic life support and advanced airway support with medical professionals, the results of the study would be different?

In the description of the participants, he did not include variables such as pathological antecedents. Do you have any facts on this?

As it says in the limitation section, there is no clear criterion to indicate the transfer. Is it possible that the start of the transfer of patients was already different taking into account the expected transfer times?

They have taken into account that there could be differences depending on the day of the week or the schedule. It is known that there are times of the day when there is greater circulatory collapse and this could influence the results of the study.

Author Response

Q) In the "study setting" you say: "The EMS system is fire department based, and certified EMS providers can provide basic and advanced life support in the airways under direct medical advice." Can you explain what "direct medical advice and support" means? Is it by phone? Do you think that if they provided basic life support and advanced airway support with medical professionals, the results of the study would be different?

A) All direct medical directions are done over the phone. After receiving a report, doctor will decide on type of advanced airways, whether to provide only BLS and start transfer depending on the situation and the level of EMS providers. However, doctor does not instruct the process of procedure, so direct medical advice may not affect the success of the procedure performed by the EMS providers. In addition, previous study has shown that prehospital advanced airways were not associated with survival, therefore, we do not think there would be a significant difference between basic life support and advanced airway support with direct medical direction. added in page 13, line 276-283)

Q) In the description of the participants, he did not include variables such as pathological antecedents. Do you have any facts on this?

A) We added information about patient’s past medical history to the patient characteristics (Table 1), but we did not include in the final analysis as they were not considered as covariates.

Q) As it says in the limitation section, there is no clear criterion to indicate the transfer. Is it possible that the start of the transfer of patients was already different taking into account the expected transfer times?

A) According to the standard protocols for 119 emergency medical service in Korea, EMS providers should provide CPR on scene at least 5 minutes. After 5 minutes, they could decide when to start transfer the patient. When making a decision, they could get information from the dispatch center about hospitals that can be transferred in the area. The median transport time interval was 7 minutes (interquartile range 4-10) in this study, and there was no significant difference between prehospital ROSC group and without prehospital ROSC group. Whether the EMS provider’s or the doctor’s decision on departure from the field were affected by the expected transport times has not been studied yet, an issue that we believe requires further research. However, in our study, we showed that when the TTI is > 10 minutes, the longer STI was negatively associated with the neurological outcome in rOHCA patients. Therefore, our study may be a basis for considering the initiation of the transfer along with the expected transfer times. (added in page 10, line 239-247)

Q) They have taken into account that there could be differences depending on the day of the week or the schedule. It is known that there are times of the day when there is greater circulatory collapse and this could influence the results of the study.

A) Information on weekdays, weekends, and holidays is additionally summarized in Table 1. These variables were not significantly different between survival to discharge and non-survival discharge groups, therefore, we did not include in the final analysis.

Reviewer 2 Report

Park HA et al performed a retrospective registry analysis to report the association between the duration of resuscitation for out of hospital cardiac arrest, transportation time, and outcomes. 

The study is interesting and the authors should be congratulated for doing it. 

A few things to improve:

  • The study results are not easy to follow. This section should be revised. Maybe provide a diagram of the analysis and results
  • The authors performed a logistic regression to control for possible confounding. However, the overall number of patients in the study is small, and many variables have small number of patient. This makes the results of logistic regression not reliable. This is supported by wide confidence intervals. This should be at least mentioned in the limitation.
  • Why were some patients transported before ROSC? 
  • This study is hypothesis generating and authors should tone down their conclusion. Authors should use the word "may" in their conclusion and should recommend larger studies/trials and not change in practice.

Author Response

Q) The authors performed a logistic regression to control for possible confounding. However, the overall number of patients in the study is small, and many variables have small number of patient. This makes the results of logistic regression not reliable. This is supported by wide confidence intervals. This should be at least mentioned in the limitation.

A) Some variables show a wide confidence interval due to the small number of patients, so multivariable logistic regression results may not be reliable. We tested goodness of fit (GOF) for survival discharge model and good neurological outcome model. In survival discharge model, when we added administration of epinephrine, amiodarone then p-value of GOF was 0.68, and the p-value was 0.91 in the model without variables. Also, in the good neurological outcome model, p-value of GOF was 0.30 with administration of epinephrine, amiodarone included, and 0.75 for the model without these variables. Therefore, we excluded the variables in the multivariable logistic regression model and Table 3 was modified. Still, in the case of endotracheal intubation and prehospital ROSC, a wide confidence interval is shown. Although, the study involved long-term data collection from multiple centers, the number of patients was not sufficient. We believe that further studies with larger patient populations are needed in the future. (added in page 13, line 284-287)

Q) Why were some patients transported before ROSC?

A) In Korea, EMS provider cannot declare death alone at the scene, so they must receive medical advice. Unless there is obvious evidence of death, all patients must be transferred to the hospital where the doctor can declare the death. Therefore, to be precise, some patients who do not show definite ROSC are not transported before ROSC, but rather transported while ROSC is undetermined or dead. In other words, all OHCA patients whose death is not clear are transported to hospitals after on scene CPR attempts. (added in page 3, line 94-98)

Q) This study is hypothesis generating and authors should tone down their conclusion. Authors should use the word "may" in their conclusion and should recommend larger studies/trials and not change in practice.

A) We changed the paragraph: In conclusion, for rOHCA patients with a shockable rhythm, continuing CPR for more than 15 minutes on the scene was associated with a decreased chance of survival and good neurological outcome. In particular, we found that in the patients whose TTI was > 10 minutes, the longer STI was negatively associated with the neurological outcome. Our study may be a basis for future studies that consider the initiation of the transfer along with the expected transfer times. (added in page 13, line 290-294)

Reviewer 3 Report

Thank you for the opportunity to review this paper. The topics are really important in terms of resuscitation science. I recommend addressing some concerns to strengthen the paper. I have the following comments.

  1. Please consider adding the definition of ROSC. How were the patients with rearrest delt with?

  1. Both primary and secondary outcomes would be markedly affected whether or not ROSC had been obtained at the scene. As a subgroup analysis, I recommend reanalyzing the data via a multivariable logistic regression model, stratifying the patients without ROSC during the STI. Basically, it is no doubt that the patients with ROSC at the scene would have shorter STI.

  1. Supplementary Figure: Please consider presenting the data with 95% CI.

  1. The description of “STI groups” in line 178 is confusing. Does this mean STI < 15 minutes?

Author Response

Q) Please consider adding the definition of ROSC. How were the patients with rearrest delt with?

A) As Utstein definition, ROSC was defined as evidence of a palpable pulse or a measurable blood pressure. The prehospital ROSC was considered as any ROSC achieved at any point during resuscitation, even if was not sustained until the patient arrived at the ED (added in page 3, line 127-130)

Q) Both primary and secondary outcomes would be markedly affected whether ROSC had been obtained at the scene. As a subgroup analysis, I recommend reanalyzing the data via a multivariable logistic regression model, stratifying the patients without ROSC during the STI. Basically, it is no doubt that the patients with ROSC at the scene would have shorter STI.

A) Previous study already shown that OHCA patients with prehospital ROSC have higher probability of survival than without prehospital ROSC. When we reanalyzed the multivariable logistic regression model without prehospital ROSC patients, STI was not significantly associated with survival to discharge (aOR, 0.45; 95% CI, 0.18–1.16). However, STI ≥ 15 minutes was still significantly negatively associated with good neurological outcome (aOR, 0.38; 95% CI, 0.22–0.67) (Supplementary Table). In patients who do not have prehospital ROSC, reducing STI and rapidly transporting them to the ED for further treatment, such as ECLS, may perhaps be associated with good neurological outcome. In particular, it may be more appropriate to perform an analysis without excluding patients with ROSC, because our study was aimed at determining whether shorter STI for rOHCA patients is associated with more favorable outcomes than waiting for ROSC in the field. (added in page 11, line 254-262)

Q) Supplementary Figure: Please consider presenting the data with 95% CI.

A) We added 95% CI bands in the Supplementary Figure.

Q) The description of “STI groups” in line 178 is confusing. Does this mean STI < 15 minutes?

A) We changed the phrase as “STI ≥ 15 minutes was significantly associated with better survival to discharge, and good neurological outcome rates”. 

Round 2

Reviewer 2 Report

None

Reviewer 3 Report

The authors have addressed all my queries appropriately. I would say the paper is acceptable in a current form.

This manuscript is a resubmission of an earlier submission. The following is a list of the peer review reports and author responses from that submission.